# The effects of visual stimulation on the cortical activity of brainstem stroke dysphagia patients: A functional near-infrared spectroscopy study

**Dandan Zhao, Yancun Li, Keyi Ning, Bingjie Zou, Bin Wang**[ID]**, Libo Li, Qiaojun Zhang**\*, **Yanping Hui**[ID]\*

Department of Rehabilitation Medicine, The Second Affiliated Hospital, Xi'an Jiaotong University, Xi'an, China

\* huiyanping@mail.xjtu.edu.cn (YH); zhangqj@mail.xjtu.edu.cn (QZ)

## Abstract

### Objective

To investigate the difference in cortical activity under food visual stimulation between patients with brainstem stroke dysphagia and healthy adults by using functional near-infrared spectroscopy (fNIRS). Additionally, we seek to identify any potential relationship between cortical activity and swallowing.

### Methods

30 patients with dysphagia after brainstem stroke and 16 healthy adults were selected. The fNIRS was used to assess the functional connection strength of global and ROIs brain regions at rest, as well as the mean change in oxygenated hemoglobin concentration ($\Delta HbO_2$) during the food visual stimulation task in both groups.

### Results

In the resting state, the functional connection strength of healthy adults ($\overline{X} = 0.514,\ s = 0.021$) was higher than that of patients ($\overline{X} = 0.472,\ s = 0.009$) ($P < 0.05$). In comparison, functional connectivity in the ROI brain region was enhanced in the patient group compared to the healthy adult group. In the task state, the patient's $\Delta HbO_2$ concentration in the left Frontopolar area, right Frontopolar area, left Orbitofrontal area and left Dorsolateral prefrontal cortex dramatically decreased in comparison to the healthy adult group. The correlation analysis revealed a moderate negative correlation between SSA and the MMSE score, VAS score, and the average $\Delta HbO_2$ concentrations in specific brain regions, including the right Frontopolar area, left Frontopolar area, left Orbitofrontal area, and left DLPFC. Furthermore, the VAS scores exhibited a moderate positive correlation with the average $\Delta HbO_2$ concentrations in the right Frontopolar area, left Frontopolar area, left Orbitofrontal area, and left DLPFC.

**Data availability statement:** All relevant data are within the paper and its Supporting information files. Data is also available from Mendeley Data. The reference link is as follows: Zhao, Dan-dan (2025), "Research Data (The effects of visual stimulation on the cortical activity of brainstem stroke dysphagia patients: A functional near-infrared spectroscopy study)", Mendeley Data, V1, doi: 10.17632/ptn3twsf59.1.

**Funding:** This research has received two grants: National Natural Science Foundation of China (81902291) and Natural Science Foundation of Shaanxi Province (2024JC-YBMS-656).

**Competing interests:** The authors have declared that no competing interests exist.

## Conclusion

Patients with brainstem stroke dysphagia showed reduced activity during visual stimulation in the Frontopolar region, the left Orbitofrontal area, and the left Dorso-lateral prefrontal cortex as compared to healthy individuals. The overall strength of functional connections was decreased, while the ROI between different brain areas increased. Following a brain stem stroke, all of these might be related to pre-oral swallowing issues.

## 1 Introduction

Post-stroke dysphagia (PSD) is commonly observed during the stroke phase, affecting a significant proportion of patients (40%−78%) [1]. It may result in severe consequences such as aspiration pneumonia, nutritional metabolism disorders, water and electrolyte imbalances, and even death [1–4]. PSD has been linked to longer hospital stays and higher death rates three months following stroke, negatively impacting patients' quality of life and rehabilitation [1]. So restoring swallowing function in PSD patients as soon as possible is of great significance. The brain stem, particularly the pons and medulla, is often regarded as essential for the swallowing process. The medulla houses the core pattern generator for swallowing, which regulates the rhythmic pattern of swallowing activity and is closely related to motor preparation and sensory processing [5]. The pons is a sensory relay system that transfers information from peripheral oral, pharyngeal, and laryngeal receptors to the upper cortex [6]. Research has indicated that patients with brain stem stroke have a more serious risk of dysphagia and aspiration than those with supratentorial stroke [7]. However, the current therapeutic treatment techniques for swallowing difficulties after brain stem stroke primarily focus on enhancing the oral phase function (e.g., delayed swallowing initiation) and the pharyngeal phase function (e.g., diminished swallowing response, aspiration, cricopharyngeal muscle atonia). Nevertheless, there is very limited research on pre-oral function, which maybe due to the challenges associated with quantifying pre-oral function. Except for the Penetration-Aspiration Scale (PAS), no sensitive index has yet been discovered to evaluate the severity of swallowing difficulty in brain stem stroke patients. Consequently, it is important to investigate the rehabilitation of swallowing function at the pre-oral phase in patients with dysphagia after brainstem stroke, so as to promote the whole-cycle rehabilitation of swallowing function.

Swallowing is a complex sensorimotor process [8]. The procedure consists of five phases: pre-oral, oral preparation, oral, pharyngeal, and esophageal. The pre-oral phase, commonly referred to as the cognitive stage, is a process in which the brain is exposed to senses including vision, hearing, and smell. Cortical and subcortical brain regions that are related to this process are then engaged, resulting in the desire to taste or the inability to try. Compared with other stages of swallowing, the pre-oral stage is more susceptible to cognitive influence. However, the relevant mechanism has not been explored. Previous neuroimaging studies have found that food visual

stimulation triggers brain responses in the visual information processing area, anterior cingulate cortex, and prefrontal cortex, all of which are involved in cognition, attention, motivation, reward processing, and decision-making [9–12]. It is vital to recognize changes in cortical excitability in this region after brain stem stroke and promote early remodeling of the swallowing-related cortical region in order to provide accurate treatment of dysphagia.

fNIRS is a functional brain neuroimaging technique that identifies neural responses in the cerebral cortex based on neurovascular coupling effects. Compared to other brain neuroimaging technologies, fNIRS is mobile, simple to use, has excellent time resolution, and is free of motion and electromagnetic interference [13,14]. Furthermore, fNIRS imposes few constraints on individuals, therefore playing an essential role in the investigation of brain function in natural settings, as well as in the application to particular populations and rehabilitation medicine. This study utilized fNIRS to examine the disparity in cortical functional status between individuals with swallowing disorders following a brain stem stroke and healthy adults during food visual stimulation. Additionally, it aimed to investigate the relationship between cortical activation and swallowing function, thus providing a theoretical foundation for the potential use of food visual cue stimulation as a rehabilitative intervention for patients with swallowing disorders after a brain stem stroke.

## 2 Materials and methods

### 2.1 Participants

The participants were 30 brain stem stroke patients with dysphagia and 16 healthy adults, who were recruited from the Department of Rehabilitation Medicine, the Second Affiliated Hospital of Xi 'an Jiaotong University from July 15, 2024 to September 18, 2024. The study procedure was approved by the Medical Ethics Committee of the Second Affiliated Hospital of Xi 'an Jiaotong University (Number: 2023474, ChiCTR2400086793), and informed consent was obtained from all participants before inclusion. All subjects have signed written informed consent.

Participants were selected according to the following criteria: Patient inclusion criteria: (1) first stroke with brain stem location, confirmed by computer tomography (CT) or magnetic resonance imaging (MRI); (2) patients with dysphagia confirmed by fluoroscopy of swallowing; (3) patients with stable vital signs, whose disease duration is between 1 and 12 months and their ages range from 18 to 85 years; (4) The patient's (MMSE) score: illiterate >17; Primary school >20 points; Middle school >23 points; University >24 points, which could cooperate with the completion of rehabilitation treatment and evaluation; (5) patients had no visual or hearing impairment.

Inclusion criteria for healthy people: (1) subjects have no neurological or psychiatric disorders, and no complaints or signs of dysphagia; (2) subjects had no visual or hearing impairment and their ages range from 18 to 85 years.

Exclusion criteria: (1) there is a history of swallowing dysfunction caused by other factors, such as Parkinson's disease, dementia, and motor neuron disease, and so on; (2) patients suffering from severe cognitive impairment or aphasia who could not continue to cooperate with treatment; (3) participants had missing skulls.

### 2.2 Clinical scale evaluation

The main clinical evaluations consist of the Standardized Swallowing Assessment (SSA), Mini-mental State Examination (MMSE), Simplified Nutritional Appetite Questionnaire (SNAQ), and Visual Analogue Scale (VAS). The swallowing function was evaluated with the SSA, cognitive function was measured with the MMSE, and hunger and food choice were examined using the SNAQ and VAS scales. Meanwhile, a fNIRS system is employed to detect alterations in brain hemodynamics while performing a task.

### 2.3 fNIRS data acquisition and swallowing task

This work used dual-wavelength near-infrared spectroscopy (730/850nm, NirScan system, Danyang Huilang, China) to collect resting and task-state brain oxygen metabolism signals (HbO$_2$/ HbR concentration variations) and dynamically monitor neurovascular coupling in the prefrontal cortex. The device has 16 avalanche photodiode (APD) detectors and

15 LED light sources, and the detection array adheres to the International Society for Optical Engineering (SPIE) standard, resulting in 48 effective detection channels. The system samples at 10 Hz, demodulates the signal using the revised Beer-Lambert equation, and topologically maps the prefrontal area using 3D spatial registration. Based on previous studies, the main areas examined in this study included the Dorsolateral prefrontal cortex (DLPFC), Middle temporal gyrus, Superior temporal gyrus, Temporopolar area, Inferior prefrontal gyrus, Frontopolar area, Orbitofrontal area, Broca's area, Pre-Motor and Supplementary Motor Cortex, Subcentral area.

The channels cover 6 regions of Interest (ROIs) on both sides of the brain: ① the Frontopolar cortex (left:11, 27, 30, 43/ right:6, 8, 21, 23, 26, 40/ middle:25, 28); ② Dorsolateral prefrontal cortex (left:32, 44, 45, 47/ right:22, 24, 38, 39, 41, 42); ③ Pre-Motor and Supplementary Motor Cortex (left:48/ right:20); ④Broca's area (left:31, 46/ right:19, 37); ⑤ Orbitofrontal area (left:9, 10/ right:7; ⑥ Inferior prefrontal gyrus (left:29/ right:4, 5). Fig 1 illustrates the arrangement of channels in the fNIRS probe set. The obtained coordinates were then converted into MNI coordinates and projected onto the Broadmann Talairach template using NirSpace's spatial registration technique.

The participants were told two hours before the experiment to avoid eating and drinking and to stay relaxed once they entered the fNIRS test room. Unnecessary movements and facial expressions were avoided as much as possible during the whole test. The experiment comprises two components: resting state and task state acquisition. The resting state was

**(a)   Location map of the brain area**

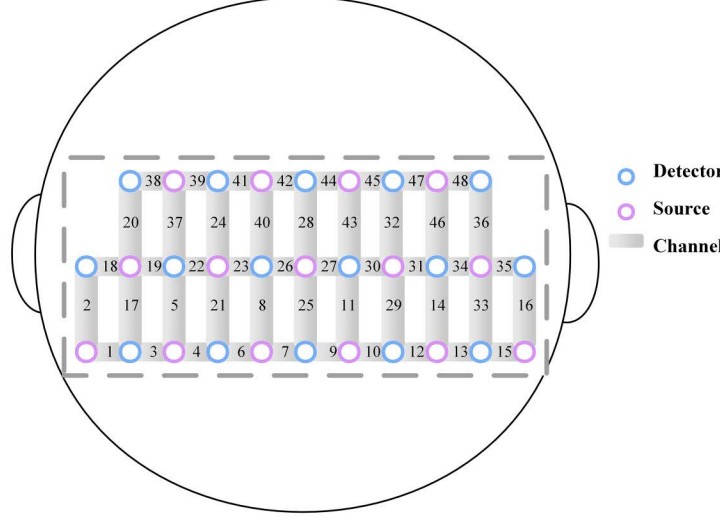

**(b)   Resting state**

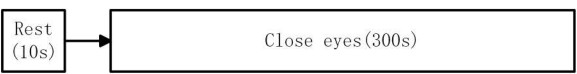

**(c)   Task state**

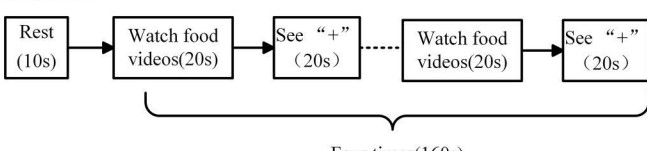

**Fig 1.  Location map of the brain area and experimental procedure.**

collected for 5 minutes, and the patients were asked to close their eyes and rest. The task approach utilized in this study consisted of two manipulation phases: 20 seconds of watching the cross and 20 seconds of watching the food video. Repeat these two phases all four times, and the total procedure took approximately 2min50s, as shown in Fig 1. The subjects were asked to avoid swallowing as much as possible. After that, the subjects were asked to fill out the Visual Analog Scale (VAS) to assess their preference for the food video.

## 2.4 fNIRS data analysis

We used the NirSpark software to preprocess and analyze the fNIRS data in our investigation. The data preprocessing involved the following steps: Spline interpolation is employed to remove motion artifacts. Band-pass filtering is applied to optical intensity signals within the frequency range of 0.01~0.10 Hz to minimize the effects of signal interference, baseline drift, and physiological noise [13]. The modified Beer-Lambert law is applied to convert light intensity into relative concentrations of $HbO_2$ and HbR. The block average result was the average of the four-block paradigms of the task-state food video viewing block paradigm $HbO_2$ minus the baseline $HbO_2$, with the increase representing the specific activation of the corresponding region. The resting state data was analyzed in block mode, with the baseline set to (-2s, 0s). The average $HbO_2$ of each channel was calculated. By dividing the sum of the $HbO_2$ concentrations of all channels in each ROI by the number of channels in each ROI, the average $HbO_2$ in the resting state of each ROI is obtained.

## 2.5 Statistical analysis

All statistical analyses were performed using SPSS 26.0 software. The individual's age and other general clinical data were represented by ($\overline{X} \pm S$). Since the sample size was less than 50, exploratory data analysis and the Shapiro-Wilk test were conducted to determine the normality of the data distribution. The independent sample t-test was employed to compare the groups. Data that deviate from the normal distribution are expressed as the median [M ($Q_L$, $Q_U$)], using the Mann-Whitney U nonparametric test. The average change in $\Delta HbO_2$ was respectively compared between the patients with swallowing disorder after brain stem stroke and healthy adults in both the resting state and task state using an independent sample t-test. Spearman linear regression was used to analyze the correlation between the average $\Delta HbO_2$ and various clinical indicators in brain stem patients with swallowing disorders after stroke and healthy adults. The strength of the correlation coefficient (r) is defined as: 0.3~0.5 indicates a weak correlation, 0.5~0.7 indicates a moderate correlation, 0.7~0.9 indicates a strong correlation, and 0.9~1.0 is an extremely high correlation. $P<0.05$ implies that there is a statistically significant difference [15].

   2.5.1 Sample size estimation. According to the literature report, the main outcome index of functional connection strength in normal subjects was $0.485 \pm 0.148$, that in the pre-experiment was $0.453 \pm 0.310$ in the normal subjects and $0.224 \pm 0.146$ in the patient group [16]. The difference test was set as bilateral $\alpha = 0.05$, power $= 0.8$. The calculated sample size is 42.

# 3 Result

## 3.1 Baseline characteristics of the subjects

In this study, we recruited a total of 30 patients with swallowing disorders after brain stem stroke and 16 healthy adults. Table 1 shows the age and scores for MMSE, VAS, and SNAQ of the subjects. There was no significant difference in age, VAS, and SNAQ baseline characteristics between groups ($P>0.05$). The MMSE scores differed considerably between the two groups ($P<0.05$).

## 3.2 Cortical activation analysis of fNIRS measurements

During the task, the $\Delta HbO_2$ concentration of the patient decreased significantly in several areas compared to the healthy adult group. These areas include the left Frontopolar area ($T=3.075$, $P=0.043$), the right Frontopolar area ($T=3.092$, $P=0.043$), the left Orbitofrontal area ($T=3.109$, $P=0.043$), and the left DLPFC ($T=3.431$, $P=0.043$). These findings

**Table 1. Baseline demographic and clinical characteristics.**

| Characteristic | HC group (N = 16) | Patient group (N = 30) | *T/Z*-value | *P*-value |
|---|---|---|---|---|
| Age ($\bar{x} \pm s$ years) | 58.9 ± 5.9 | 61.9 ± 11.9 | 1.1387[a] | 0.173 |
| MMSE [$M(P_{25}, P_{75})$] | 30.0 (29.0, 30.0) | 28.0 (26.3, 30.0) | −2.684[b] | 0.007* |
| VAS ($\bar{x} \pm s$) | 8.70 ± 1.1 | 8.58 ± 1.8 | 0.242 | 0.81 |
| SNAQ ($\bar{x} \pm s$) | 16.5 ± 1.4 | 16.0 ± 2.8 | 0.702 | 0.486 |

MMSE, mini-mental state examination; VAS, Visual Analogue Scale; SNAQ, Simplified Nutritional Appetite Questionnaire; [a] is the *T* value and [b] is the *Z* value.

suggested that, in response to dynamic food cues, there was less activation in the frontopolar area, left orbitofrontal area, and left DLPFC than in healthy individuals. See Figs 2 and 3 for further information, and Table 2 for particular data.

In the resting state, the functional connectivity strength in the healthy adults' group ($\bar{X} = 0.514$, $s = 0.021$) was significantly higher than that in the patients' group ($\bar{X} = 0.472$, $s = 0.009$) ($P<0.05$), indicating that there were more connections between brain regions in the healthy adults at rest compared to the patients, as shown in Fig 4A and B. The P-values for functional connectivity in six ROI locations of patients and healthy people are displayed in Fig 4C. In comparison, the patient group showed increased functional connectivity between the right Broca's area and the Frontopolar area (left: $T=−2.768$, $P=0.046$; right: $T=−3.820$, $P=0.013$; middle: $T=−3.376$, $P=0.014$), as well as between the DLPFC and the right and middle Frontopolar areas (right: $T=−3.126$, $P=0.024$; middle: $T=−3.582$, $P=0.010$). Additionally, enhanced connectivity was observed between the Pre-motor and supplementary motor cortex and the right and middle Frontopolar areas in the patient group (right: $T=−3.504$, $P=0.011$; middle: $T=−2.937$, $P=0.032$). The functional connectivity between the right Inferior frontal gyrus and the Dorsolateral prefrontal cortex (left: $T=−2.970$, $P=0.032$; right: $T=−4.908$, $P=0.000$), Frontopolar area (left: $T=−4.024$, $P=0.004$; right: $T=−4.851$, $P=0.000$; middle: $T=−4.769$, $P=0.000$), and Orbitofrontal area was also strengthened (left: $T=−4.793$, $P=0.000$; right: $T=−3.106$, $P=0.024$). In the task state, the patient's $\Delta HbO_2$ concentration in the left Frontopolar area, right Frontopolar area, left Orbitofrontal area and left Dorsolateral prefrontal cortex dramatically decreased in comparison to the healthy adult group. These findings are summarized in Table 3.

### 3.3 Correlation analysis

A correlation study was conducted to investigate the association between the clinical markers SSA score and MMSE score, VAS score, average $\Delta HbO_2$ concentration in the left Frontopolar area, right Frontopolar area, left Orbitofrontal area, and left DLPFC of the participants in the two groups. The results indicated a significant negative association between the SSA score and both the MMSE score ($r=−0.639$, $P=0.000$) and the VAS score ($r=−0.602$, $P=0.000$). Furthermore, a weak negative connection was found between the SSA score and the $\Delta HbO_2$ concentrations in specific brain areas. These areas include the left Frontopolar area ($r=−0.431$, $P=0.003$), right Frontopolar area ($r=−0.400$, $P=0.006$), left Orbitofrontal area ($r=−0.391$, $P=0.000$), and left DLPFC ($r=−0.367$, $P=0.012$). According to the results, worse cognition, decreased appetite, and reduced brain activity in certain brain regions (Frontopolar area, left Orbitofrontal area, and left DLPFC) are associated with a higher chance of aspiration, as illustrated in Fig 5.

Ultimately, the VAS score was associated with the average $\Delta HbO_2$ concentration in specific brain regions. The results demonstrated a significant positive correlation between the VAS score and the left Frontopolar area ($r=0.528$, $P=0.000$), the right Frontopolar area ($r=0.426$, $P=0.003$), the left orbitofrontal area ($r=0.374$, $P=0.000$), and the left DLPFC ($r=0.498$, $P=0.000$). Fig 6 suggests that increased brain activity in the Frontopolar area, left Orbitofrontal area, and left DLPFC is linked to improved appetite performance.

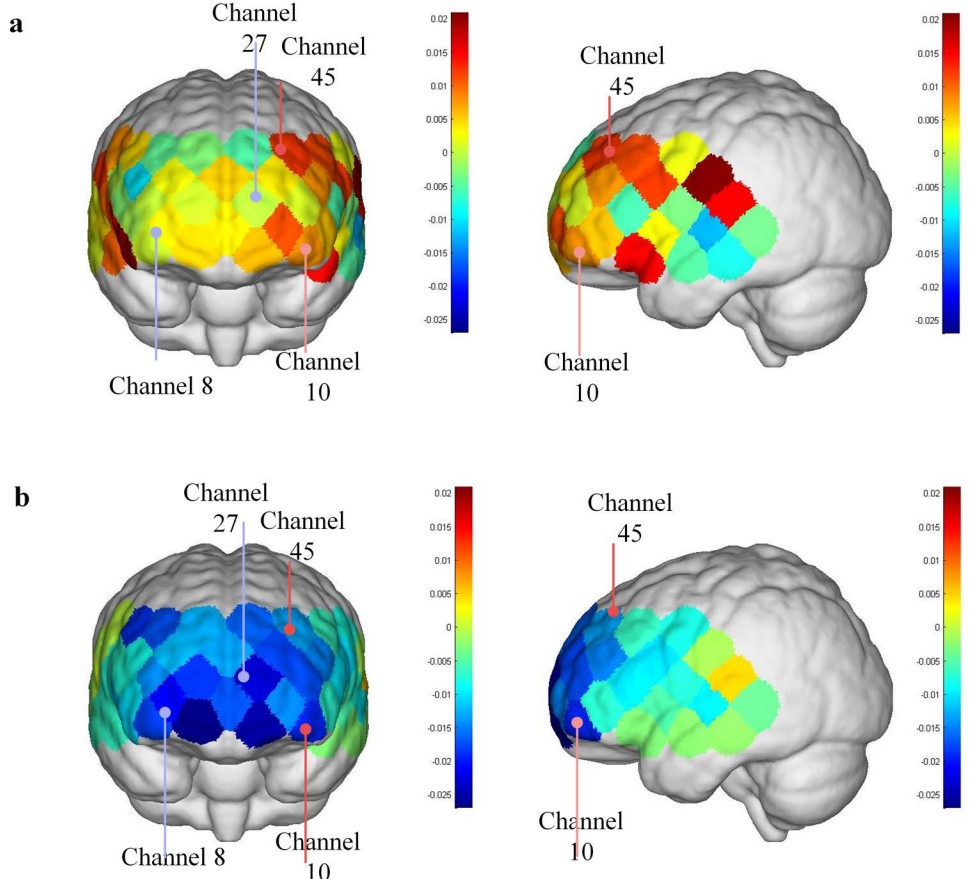

**Fig 2. Topographic map of ΔHbO$_2$ concentration in the task state.**

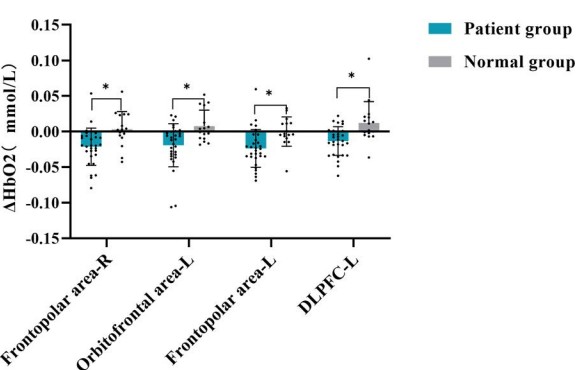

**Fig 3. Comparison of average ΔHbO$_2$ concentration between normal and patients in task state.**

## 4 Discussion

In this study, patients with dysphagia after brain stem stroke and healthy adults were chosen as research subjects. fNIRS was used to explore differences in cortical functional activities between healthy adults and patients under food visual stimulation. The correlation between the mean value of HbO$_2$ and SSA, MMSE, VASc scales was analyzed.

**Table 2. Intergroup comparison of brain area activation in food visual stimulation tasks.**

| Channel | Label of SD | Brain region | T-value | P-value |
|---|---|---|---|---|
| 1 | S1-D1 | Middle Temporal gyrus-R | 0.641 | 0.681 |
| 2 | S1-D6 | Superior Temporal Gyrus-R | 0.436 | 0.798 |
| 3 | S2-D1 | Temporopolar area-R | 1.331 | 0.315 |
| 4 | S2-D2 | Inferior prefrontal gyrus-R | 2.194 | 0.124 |
| 5 | S2-D7 | Inferior prefrontal gyrus-R | 2.037 | 0.136 |
| 6 | S3-D2 | Frontopolar area-R | 2.355 | 0.097 |
| 7 | S3-D3 | Orbitofrontal area-R | 2.575 | 0.081 |
| 8 | S3-D8 | Frontopolar area-N | 3.092 | 0.043* |
| 9 | S4-D3 | Orbitofrontal area-L | 2.658 | 0.075 |
| 10 | S4-D4 | Orbitofrontal area-L | 3.109 | 0.043* |
| 11 | S4-D9 | Frontopolar area-L | 2.803 | 0.056 |
| 12 | S5-D4 | Temporopolar area-L | 1.696 | 0.222 |
| 13 | S5-D5 | Middle Temporal gyrus-L | −0.263 | 0.866 |
| 14 | S5-D10 | Superior Temporal Gyrus-L | 0.469 | 0.798 |
| 15 | S6-D5 | Middle Temporal gyrus-L | −0.391 | 0.798 |
| 16 | S6-D11 | Middle Temporal gyrus-L | 0.047 | 0.963 |
| 17 | S7-D1 | Superior Temporal Gyrus-R | 1.037 | 0.473 |
| 18 | S7-D6 | Subcentral area-R | 0.120 | 0.925 |
| 19 | S7-D7 | Broca's area-R | 1.498 | 0.261 |
| 20 | S7-D12 | Pre-Motor and Supplementary Motor Cortex-R | 1.540 | 0.261 |
| 21 | S8-D2 | Frontopolar area-R | 0.702 | 0.648 |
| 22 | S8-D7 | Dorsolateral prefrontal cortex-R | 0.393 | 0.798 |
| 23 | S8-D8 | Frontopolar area-R | 1.334 | 0.315 |
| 24 | S8-D13 | Dorsolateral prefrontal cortex-R | −0.166 | 0.906 |
| 25 | S9-D3 | Frontopolar area-N | 1.999 | 0.136 |
| 26 | S9-D8 | Frontopolar area-R | 2.344 | 0.097 |
| 27 | S9-D9 | Frontopolar area-L | 3.075 | 0.043* |
| 28 | S9-D14 | Frontopolar area-N | 2.388 | 0.097 |
| 29 | S10-D4 | Inferior prefrontal gyrus-L | 1.372 | 0.315 |
| 30 | S10-D9 | Frontopolar area-L | 1.759 | 0.205 |
| 31 | S10-D10 | Broca's area-L | 0.459 | 0.798 |
| 32 | S10-D15 | Dorsolateral prefrontal cortex-L | 2.334 | 0.097 |
| 33 | S11-D5 | Middle Temporal gyrus-L | −0.225 | 0.878 |
| 34 | S11-D10 | Superior Temporal Gyrus-L | 0.291 | 0.863 |
| 35 | S11-D11 | Primary and Auditory Association Cortex-L | 0.737 | 0.648 |
| 36 | S11-D16 | Subcentral area-L | 2.105 | 0.136 |
| 37 | S12-D7 | Broca's area-R | 0.719 | 0.648 |
| 38 | S12-D12 | Dorsolateral prefrontal cortex-R | 0.964 | 0.511 |
| 39 | S12-D13 | Dorsolateral prefrontal cortex-R | 2.891 | 0.057 |
| 40 | S13-D8 | Frontopolar area-R | 1.498 | 0.261 |
| 41 | S13-D13 | Dorsolateral prefrontal cortex-R | 1.599 | 0.244 |
| 42 | S13-D14 | Dorsolateral prefrontal cortex-R | 1.286 | 0.328 |
| 43 | S14-D9 | Frontopolar area-L | 2.023 | 0.136 |
| 44 | S14-D14 | Dorsolateral prefrontal cortex-L | 2.020 | 0.136 |
| 45 | S14-D15 | Dorsolateral prefrontal cortex-L | 3.431 | 0.043* |
| 46 | S15-D10 | Broca's area-L | 1.669 | 0.223 |

*(Continued)*

**Table 2.** (Continued)

| Channel | Label of SD | Brain region | T-value | P-value |
|---------|-------------|--------------|---------|---------|
| 47 | S15-D15 | Dorsolateral prefrontal cortex-L | 1.980 | 0.136 |
| 48 | S15-D16 | Pre-Motor and Supplementary Motor Cortex-L | 0.712 | 0.648 |

## 4.1 Features of prefrontal activation during food visual stimulation tasks

In this study, fNIRS was used to detect hemodynamic alterations in the cortex of the brain, altering the research paradigm of the neurological processes of swallowing due to its millisecond temporal resolution, natural setting adaption, and demographic applicability. Furthermore, block design was adopted in fNIRS detection in this study. Based on prior research and preliminary experiments, it can make brain activation activities more obvious, which is similar to the results obtained by event-related design method in swallowing task [16,17]. In the meanwhile, it is easier to use and evaluate than event-related design, and the superposition of stimuli can result in more activation and improved detection.

This research demonstrated the neural activity features of individuals with dysphagia following brain stem stroke in the preoral time using a fNIRS technology system, which is as follows:

The average $\Delta HbO_2$ concentration in the Frontopolar area, left Orbitofrontal area, and left DLPFC in patients with food visual stimulation is lower than in healthy adults, indicating that cortical activity is suppressed, which has been mentioned in previous studies on swallowing and food visual stimulation [9,11,18,19]. Previous neuroimaging studies on food visual stimuli show that food pictures elicit a synergistic response in multiple brain functional regions, including the insular cortex, limbic system, orbitofrontal cortex, parietal region, and striatum [11,20–22]. Among them, the visual cortex and the orbitofrontal cortex of the brain responded more strongly to food cues. Furthermore, food cues engaged the prefrontal cortex, a brain area involved in attention management and reward processing, as well as the regulation of generalized arousal associated with emotional experiences. Some studies imply that the prefrontal cortex is involved in swallowing-related functions [23,24], and the visual cortex provides sensory input for all further cognitive processes [12], promoting the occurrence of eating behavior [25]. Furthermore, Barer revealed that visual impairment is linked to dysphagia in a study on stroke and persistent dysphagia [18], which is in line with the experimental findings of this investigation. As a result, we believe that food visual stimulation plays an essential part in the regulation of swallowing function, which may have a significant impact on cognitive processes such as swallowing preparation and initiation by activating the brain's perception-cognitive-motor integration network, and could be a new breakthrough in the rehabilitation of pre-oral swallowing disorders.

## 4.2 Resting state functional connection strength comparison

At rest, the healthy adult group had stronger total functional connectivity than the sick group; however, the functional connectivity of the ROI brain area improved in the patient group relative to the healthy adults. The result suggest an enhanced association between ROI brain regions in patients with dysphagia after brain stem stroke. This is inconsistent with the findings of Xu et al [26]. However, Quinlan EB et al [27]. discovered that after three weeks of standardized robotic treatment in patients with severe motor disorders, the functional connection strength between the affected motor cortex and the contralateral premotor area increased significantly more than in the mild group, implying that the severity of injury may reverse the neuroplasticity response pattern. Cai et al [28]. further support the variability of this compensatory restructuring technique. They discovered that hemiplegia patients with varying degrees of motor dysfunction had distinct functional connectivity patterns. Patients with severe hemiplegia exhibited strong connectivity in certain brain areas, indicating that motor systems may adapt for nerve injury by improving specific route connections. In addition, K.osier et al [19] reported that there exists functional connectivity among several brain regions, both intra-regionally and inter-regionally. Each brain region carries out different roles in s in the process of sensorimotor planning and execution, and the functional

a

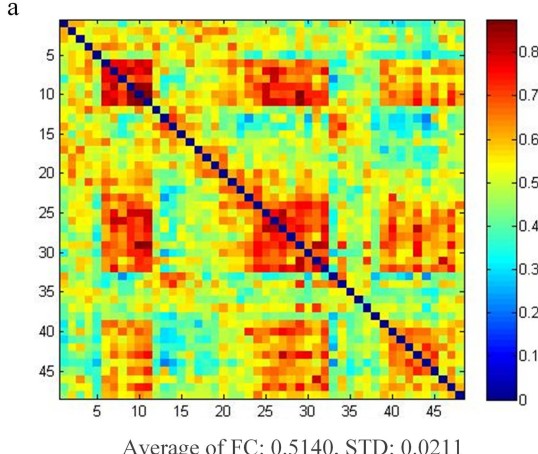

Average of FC: 0.5140, STD: 0.0211

b

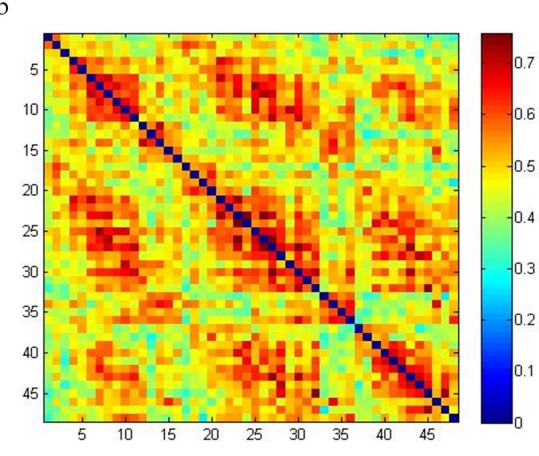

Average of FC: 0.4722, STD: 0.0915

c

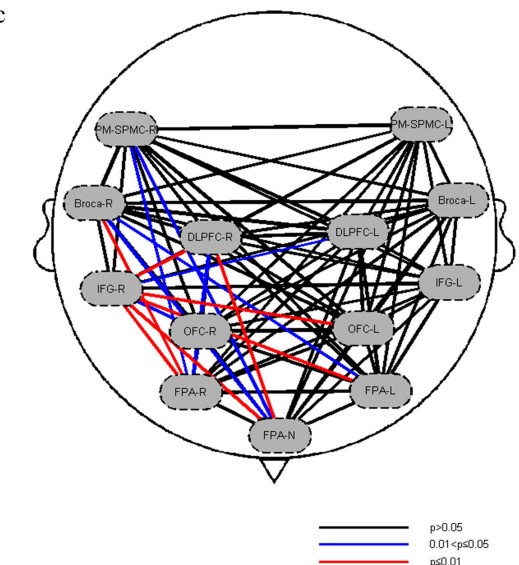

p>0.05
0.01<p≤0.05
p≤0.01

**Fig 4. Functional connection diagram.**

**Table 3. Intergroup comparison of ROI functional connectivity in resting tasks.**

| ROI Brain region | T | P |
|---|---|---|
| Broca's area-R- Frontopolar area-L | −2.767 | 0.046* |
| Broca's area-R – Frontopolar area-N | −3.376 | 0.014* |
| Broca's area-R – Frontopolar area-R | −3.820 | 0.006** |
| Dorsolateral prefrontal cortex-R – Frontopolar area-N | −3.582 | 0.010* |
| Dorsolateral prefrontal cortex-R – Frontopolar area-R | −3.126 | 0.024* |
| Inferior prefrontal gyrus-R – Dorsolateral prefrontal cortex-R | −4.908 | 0.000** |
| Inferior prefrontal gyrus-R – Dorsolateral prefrontal cortex-L | −2.970 | 0.032* |
| Inferior prefrontal gyrus-R – Frontopolar area-L | −4.024 | 0.004** |
| Inferior prefrontal gyrus-R – Frontopolar area-N | −4.769 | 0.000** |
| Inferior prefrontal gyrus-R – Frontopolar area-R | −4.851 | 0.000** |
| Inferior prefrontal gyrus-R – Orbitofrontal area-L | −4.793 | 0.000** |
| Inferior prefrontal gyrus-R – Orbitofrontal area-R | −3.106 | 0.024* |
| Pre-Motor and Supplementary Motor Cortex-R – Frontopolar area-N | −2.937 | 0.032* |
| Pre-Motor and Supplementary Motor Cortex-R – Frontopolar area-N | −3.504 | 0.011* |

connection between these regions is the result of regulating actions during swallowing tasks. Thus, we hypothesized that the enhancement of functional connectivity in patients' brain areas was correlated with the severity of swallowing disorders after brain stem stroke. In addition, there is a significant right-sided skewness in the enhancement of the correlation between the ROI brain regions of patients in the resting state. This might be because the above brain regions are recruited after injury and participate in the functional compensation of swallowing.

### 4.3 Correlation analysis of SSA, MMAE and VAS scores with mean $HbO_2$ of brain channel

The research demonstrated that the SSA score was moderately negatively correlated with the MMSE score, VAS score, and the average $\Delta HbO_2$ concentration in the Frontopolar area, the left Orbitofrontal area, and the left DLPFC. This suggests that cognitive decline, decreased appetite, and decreased brain functional activity in the Frontopolar area, left Orbitofrontal area, and left DLPFC are related with an increased risk of aspiration. In addition, VAS scores were moderately positively correlated with average $\Delta HbO_2$ concentrations in the Frontopolar area, the left Orbitofrontal area, and the left DLPFC, suggesting that increased changes in brain functional activity in these regions were positively correlated with higher appetite performance. Previous research has indicated that brain lesions can result in a decrease in cognitive function, which can subsequently impair the ability to control swallowing and lead to dysphagia [18,29–32], which researchers believe is due to the frontal cortex's involvement in complex cognitive behavior planning, decision making, and execution [18]. A substantial amount of data indicates that the cerebral cortex is crucial for the initiation and regulation of swallowing, even though the swallowing center mediates the swallowing reflex as a whole. In an animal experiment, subprimates' frontal cortexes were activated to cause swallowing [33,34]. Concurrently, the loss of PFC in domestic rabbits has been shown to induce swallowing difficulties in these animals [35]. Together with the findings of this investigation, we propose that the frontal lobe-led cognitive-motivation regulating network may play an important role in the pre-oral phase of swallowing.

Few research has been done on pre-oral dysphagia, and most papers on dysphagia have focused on the oral and pharyngeal phases. Furthermore, the preoral swallowing cortex's coordination mechanism and labor division are unclear, and the connection between dysphagia and cognition has not been adequately examined. Studies have found that changes in the functional activity of brain areas in patients with dysphagia after stroke can be used to predict the characteristics and risk of swallowing and can be used as an auxiliary means for clinical diagnosis and treatment [36]. The purpose of this research is to examine the differences in brain activation during resting and task states between healthy adults and

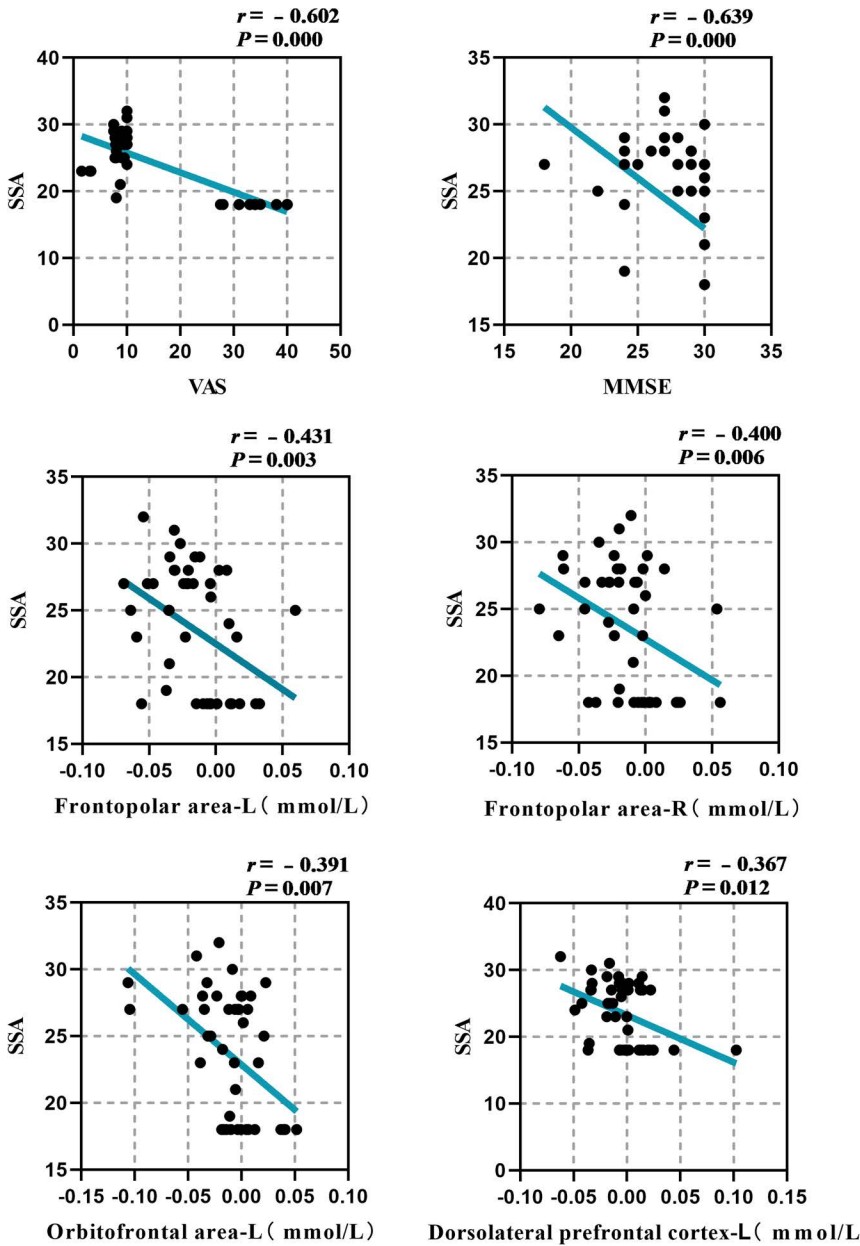

**Fig 5. Correlation analysis of the SSA score with clinical indicators and average ΔHbO$_2$ concentration of brain channel.**

patients with swallowing disorders following brain stem stroke. This will help further to understand the intricate relationship between preoral swallowing and cognition and provide additional evidence for a deeper understanding of the brain network mechanism underlying it.

### 4.4 Research limitations and future directions

Furthermore, the study was limited by the small sample size, the absence of a comprehensive swallow angiography data set to monitor preoral signs, the lack of a severity grading analysis of brainstem stroke patients, and the lack of a

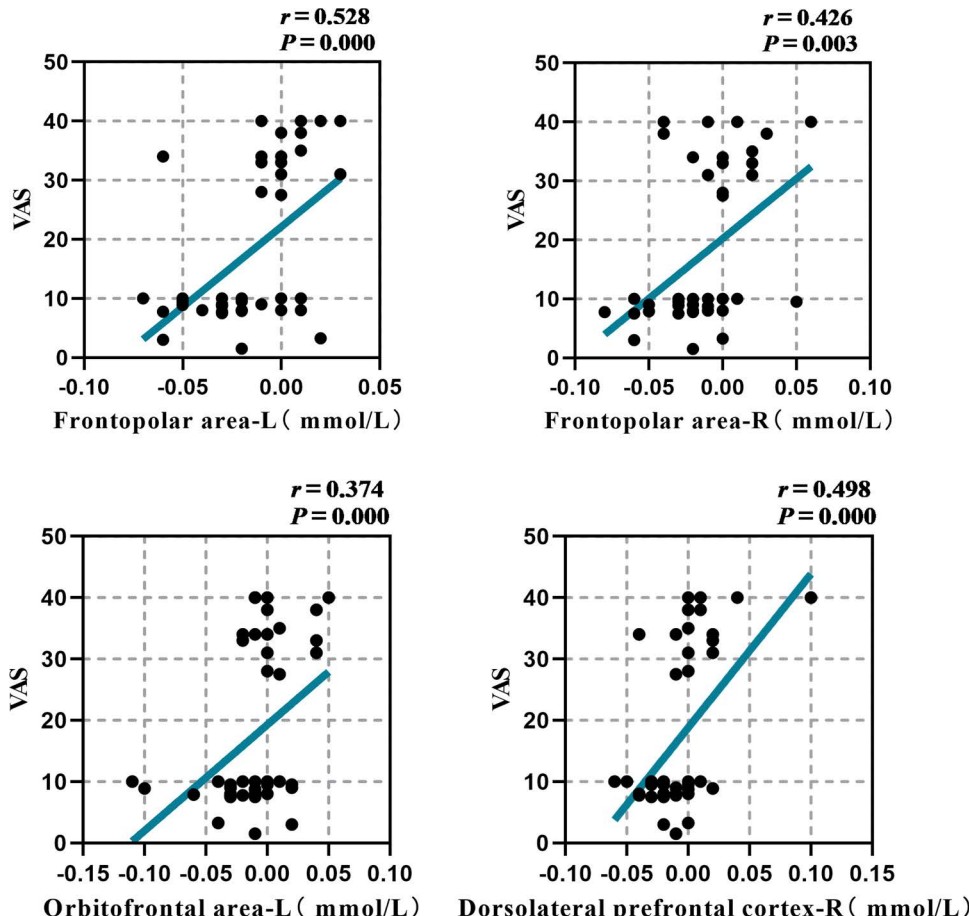

**Fig 6. Correlation analysis of the VAS score with the average ΔHbO$_2$ concentration of brain channel.**

long-term intervention about the brain areas discovered in the study. To classify swallowing difficulties according to severity, more patients will be recruited in the subsequent study. Additionally, pre-oral data from swallowing angiography and fNIRS combined analysis will be gathered to draw more precise conclusions.

## 5 Conclusion

In summary, food visual stimulation can induce alterations in the cerebral cortex, which is closely related to oral precognitive function, in patients who have swallowing disorders following brain stem stroke. The Frontopolar area, the left Orbitofrontal area and the left DLPFC area identified in this study may be therapeutic targets for clinical stimulation and may be involved in the regulation of neuroplasticity in individuals with swallowing disorders following brain stem stroke.

## Supporting information

**S1 Table. Baseline demographic and clinical characteristics.**
(DOCX)

**S2 Table. Intergroup comparison of brain area activation in food visual stimulation tasks.**
(DOCX)

**S3 Table. Intergroup comparison of ROI functional connectivity in resting tasks.**
(DOCX)

**S1 File. Research data.**
(XLSX)

## Author contributions

**Conceptualization:** Dandan Zhao, Libo Li, Qiaojun Zhang.

**Data curation:** Yancun Li, Keyi Ning, Bingjie Zou, Bin Wang.

**Formal analysis:** Dandan Zhao, Keyi Ning, Bingjie Zou, Libo Li.

**Investigation:** Dandan Zhao, Bingjie Zou.

**Project administration:** Qiaojun Zhang, Yanping Hui.

**Validation:** Qiaojun Zhang.

**Writing – original draft:** Dandan Zhao.

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
