## [Decision Letter · Decision Letter 0]

18 Mar 2025

PONE-D-24-45819The effects of visual stimulation on the cortical activity of brainstem stroke dysphagia patients: A functional near-infrared spectroscopy studyPLOS ONE

Dear Dr. Hui,

Thank you for submitting your manuscript to PLOS ONE. After careful consideration, we feel that it has merit but does not fully meet PLOS ONE’s publication criteria as it currently stands. Therefore, we invite you to submit a revised version of the manuscript that addresses the points raised during the review process.There are minor comments. Hopefully the authors will easily take care of the issues raised by reviewers.

We look forward to receiving your revised manuscript.

Kind regards,

Shashank Shekhar, MD

Academic Editor

PLOS ONE

Journal Requirements:

2. Thank you for submitting the above manuscript to PLOS ONE. During our internal evaluation of the manuscript, we found significant text overlap between your submission and previous work in the [introduction, conclusion, etc.].

Please revise the manuscript to rephrase the duplicated text, cite your sources, and provide details as to how the current manuscript advances on previous work. Please note that further consideration is dependent on the submission of a manuscript that addresses these concerns about the overlap in text with published work.

[If the overlap is with the authors’ own works: Moreover, upon submission, authors must confirm that the manuscript, or any related manuscript, is not currently under consideration or accepted elsewhere. If related work has been submitted to PLOS ONE or elsewhere, authors must include a copy with the submitted article. Reviewers will be asked to comment on the overlap between related submissions (http://journals.plos.org/plosone/s/submission-guidelines#loc-related-manuscripts) .]

We will carefully review your manuscript upon resubmission and further consideration of the manuscript is dependent on the text overlap being addressed in full. Please ensure that your revision is thorough as failure to address the concerns to our satisfaction may result in your submission not being considered further

“This work was funded by National Natural Science Foundation of China (81902291) and Natural Science Foundation of Shaanxi Province (2024JC-YBMS-656). We thank our co-workers at the The Second Affiliated Hospital of Xi 'an Jiaotong University.”

6. Thank you for stating the following financial disclosure:

Reviewers' comments:

Reviewer's Responses to Questions

**Comments to the Author**

1. Is the manuscript technically sound, and do the data support the conclusions?

Reviewer #1: Partly

Reviewer #2: Yes

2. Has the statistical analysis been performed appropriately and rigorously? 

Reviewer #1: Yes

Reviewer #2: Yes

3. Have the authors made all data underlying the findings in their manuscript fully available?

Reviewer #1: Yes

Reviewer #2: Yes

4. Is the manuscript presented in an intelligible fashion and written in standard English?

Reviewer #1: Yes

Reviewer #2: Yes

5. Review Comments to the Author

Reviewer #1: Thank you for submitting your manuscript. The aims of the paper are articulated. However, the design has certain flaws that need addressing, as detailed below:

1. The Visual Analog Scale (VAS) is generally used for pain assessment. Is it appropriate to assess dysphagia?

2. The patients' mini-mental state examination (MMSE) was≥17 points. Should people with different academic qualifications be distinguished?

3. The discussion section needs to be optimised in detail.

Reviewer #2: 1) The abstract does not highlight the focus on functional connections between different brain ROIs in the methods section.

2) The conclusion states - "The overall strength of functional connections was decreased, while the ROI between different brain areas increased." - this is quite ambiguous, and possibly misleading. I am assuming that it was meant to imply that the functional connections between specific ROIs in the brain are increased.

3) Between the patients selected who suffered a brainstem stroke and healthy patients, was there a significant difference in the resting HbO2 levels?

3) similarly, is there any clinical data to rule out the possibility of pre-existing conditions in brainstem stroke patients which may have impacted the delta HbO2 further? - such as smoking, COPD, diabetes, peripheral artery disease, carotid or vertebral artery stenosis?

Additionally, it may be helpful to give references to help negate or quantify the effect of these disorders on the levels of HbO2.

6. PLOS authors have the option to publish the peer review history of their article (what does this mean? ). If published, this will include your full peer review and any attached files.

**Do you want your identity to be public for this peer review?** For information about this choice, including consent withdrawal, please see our Privacy Policy .

Reviewer #1: No

Reviewer #2: No

---

## [Author Response · Author response to Decision Letter 1]

14 Apr 2025

No.: PONE-D-24-45819R1

Title: The effects of visual stimulation on the cortical activity of brainstem stroke dysphagia patients: A functional near-infrared spectroscopy study

First author: Dr. Dandan Zhao

Corresponding author: Prof. Yanping Hui

Dear editor and reviewers:

The authors thank you for the comments concerning our manuscript. The comments are all valuable and helpful for us to revise and improve the manuscript. We have revised the manuscript seriously according to the comments and suggestions of the reviewers. In the following we specify changes made in the manuscript, and reply point-by-point to the reviewers’ comments.

Thanks!

Prof. Yanping Hui

Response to editor

Major Comments:

Comment 1: Please ensure that your manuscript meets PLOS ONE's style requirements, including those for file naming. The PLOS ONE style templates can be found at

Response 1: Thank you very much for your reminding. In an effort to comply with PLOS One's publishing requirements, we have modified the manuscript's format in accordance with PLOS One's template. Changes were made as follows: The first page of the text had the title, author, and unit; Make use of the appropriate typeface and title format; After the first paragraph that is cited, put the title of each graph; Place the table immediately below the paragraph that was cited first; Changed the format of citations and supporting information in the manuscript; Each researcher's contributions are described at the conclusion of the study; Remove the financial disclosure statement from the text.

Comment 2�Thank you for submitting the above manuscript to PLOS ONE. During our internal evaluation of the manuscript, we found significant text overlap between your submission and previous work in the [introduction, conclusion, etc.].

Please revise the manuscript to rephrase the duplicated text, cite your sources, and provide details as to how the current manuscript advances on previous work. Please note that further consideration is dependent on the submission of a manuscript that addresses these concerns about the overlap in text with published work.

Response 2�Thank you for your feedback about the overlap in our articles. We sincerely apologize for this oversight and have reworded some of the overlapping parts of the introduction, methods, discussions and conclusions. The complete text repetition rate is less than 20%, according to the Turntin and iThenticate reviews, and the citations have been suitably appended to the earlier work. We further extend the introduction to concentrate on the pre-oral period of swallowing in contrast to earlier PSD research. Additionally, the use of near-infrared spectroscopy instruments allows individuals to collect brain function data under natural conditions to meet the needs of food visual stimulation tasks. In the discussion section, we emphasize the connection between pre-oral swallowing and cognition and highlight the benefits of the fNIRS block design in this investigation. At the same time, we demonstrate that fNIRS brain function changes may be used to predict swallowing features and risk, and can be employed as a supplementary method of clinical diagnosis and therapy, particularly given the existing challenges in measuring preoral function. Furthermore, our research established a theoretical basis for the use of food visual stimulation as a therapeutic preoral therapy method.

Comment 3: Thank you for stating the following in the Acknowledgments Section of your manuscript:

“This work was funded by National Natural Science Foundation of China (81902291) and Natural Science Foundation of Shaanxi Province (2024JC-YBMS-656). We thank our co-workers at the The Second Affiliated Hospital of Xi 'an Jiaotong University.”

Response 3�Thanks for your careful reading and helpful comments on our manuscript, we have removed the text related to the grant from the manuscript. We would like to update the funding statement for this study as follows:

“This research has received two grants: National Natural Science Foundation of China (81902291) and Natural Science Foundation of Shaanxi Province (2024JC-YBMS-656).” I included this section in the cover letter as well.

Comment 4: When completing the data availability statement of the submission form, you indicated that you will make your data available on acceptance. We strongly recommend all authors decide on a data sharing plan before acceptance, as the process can be lengthy and hold up publication timelines. Please note that, though access restrictions are acceptable now, your entire data will need to be made freely accessible if your manuscript is accepted for publication. This policy applies to all data except where public deposition would breach compliance with the protocol approved by your research ethics board. If you are unable to adhere to our open data policy, please kindly revise your statement to explain your reasoning and we will seek the editor's input on an exemption. Please be assured that, once you have provided your new statement, the assessment of your exemption will not hold up the peer review process.

Response 4�Thank you for your reminding. All the data in this study have been reflected in the manuscript and supporting information files.

Comment 5: We note that the grant information you provided in the ‘Funding Information’ and ‘Financial Disclosure’ sections do not match.

Response 5�We notice this problem and sincerely apologize for any trouble that this error may have caused you. We will ensure that the correct grant numbers is provided for the awards in the "Funding Information" section when we resubmit.

Comment 6: Thank you for stating the following financial disclosure:

Response 6�I apologize for any confusion I may have caused you. I will answer the following questions and explain them in the submission letter.

a) The research was funded by the National Natural Science Foundation of China (81902291) and the Natural Science Foundation of Shaanxi Province (2024JC-YBMS-656).

b) Funders have no role in research design, data collection and analysis, decision to publish, or preparation of the manuscript.

c)None

d) This research was supported by the National Natural Science Foundation of China (81902291) and the Natural Science Foundation of Shaanxi Province (2024JC-YBMS-656).

Comment 7: Please review your reference list to ensure that it is complete and correct. If you have cited papers that have been retracted, please include the rationale for doing so in the manuscript text, or remove these references and replace them with relevant current references. Any changes to the reference list should be mentioned in the rebuttal letter that accompanies your revised manuscript. If you need to cite a retracted article, indicate the article’s retracted status in the References list and also include a citation and full reference for the retraction notice.

Response 7: Thank you for reminding us. We have checked the references one by one to make sure they are complete and correct.

Response to reviewers

Reviewer #1

General comments: The aims of the paper are articulated. However, the design has certain flaws that need addressing

Response: Thanks to the reviewer for approving our idea. We will be very happy to edit the text further and answer your query, based on helpful comments from you.

Comments 1: The Visual Analog Scale (VAS) is generally used for pain assessment. Is it appropriate to assess dysphagia?

Thank you very much for your questions about our manuscript. The Visual Analog Scale (VAS) was used in this research to evaluate the participants' appetites. We selected this scale since prior research on eating disorders utilized VAS scale to measure the extent of participants' food cravings (Adi Ziv., 2020), and VAS scale was also used to measure the demand for drugs in drug addiction and associated research (Xiao Lin & Jiahui Deng & Kai Yuan., 2021). I thought the reason you were confused was because I did not fully describe the purpose of the VAS scale in the method section. Added modifications have been made to the original manuscript (P 7, line 132-134 from the top).

Comments 2: The patients' mini-mental state examination (MMSE) was≥17 points. Should people with different academic qualifications be distinguished?

Response 2: We really appreciate you bringing up our inadequacies. The patient's MMSE score was as follows when we included them: illiteracy >17; primary school >20 points; middle school >23 points; and university >24 points, which could cooperate with the completion of rehabilitation treatment and evaluation. Following this, the median MMSE for the patient group was 28 and for the healthy group was 30 (refer to Table 1), and all of the individuals we recruited had MMSE scores more than 24 points. However, the inclusion criteria were not explained in detail, and the inclusion criteria of this part were refined and supplemented in the original text. (P 6, line 118-120 from the top). We will also be careful to prevent such issues in the future. Thank you again for your careful review.

Comments 3: The discussion section needs to be optimised in detail.

Response 3: Thank you very much for your suggestion, we have reorganized the paragraph structure of the discussion section to make its logic clearer. In addition, we expanded the discussion section's content, explained the significance of the results in detail, fully discussed the similarities and differences with existing studies. Furthermore, we clarified the study's innovation points, limitations, and future development. (P 15, line 278-384 from the top)

Reviewer #2

Comments 1: The abstract does not highlight the focus on functional connections between different brain ROIs in the methods section.

Response 1: Thanks to your suggestion, we have supplemented the technical details of functional connections between different brain ROIs in the methods section of the abstract

The fNIRS was used to assess the functional connection strength of global and ROIs brain regions at rest, as well as the mean change in oxygenated hemoglobin concentration (ΔHbO2) during the food visual stimulation task in both groups. (P 2, line 29-31 from the top)

The selection of the ROI brain area is available in the Methods section of the text. (P 7, line 151-156 from the top)

Comments 2: The conclusion states - "The overall strength of functional connections was decreased, while the ROI between different brain areas increased." - this is quite ambiguous, and possibly misleading. I am assuming that it was meant to imply that the functional connections between specific ROIs in the brain are increased.

Response 2: I am very sorry for the confusion caused to you. In the specific analysis of the study, the overall functional connectivity of patients with swallowing disorder after brain stem stroke decreased, while the functional connectivity between specific ROIs brain regions increased, as shown in Table 2. We also reviewed some literatures related to fNIRS and fMRI to explain this result. Xu et al.'s study indicated that the average strength of functional brain network connections was higher in healthy subjects than in patients. However, intra-hemisphere or interhemisphere connections, as well as the functional connection map, were more sparse in the stroke group than in the control group (Gongcheng Xu., 2024). But according to our analysis of the literature, Cai et al. contrasted various FC (functional connectivity) patterns amongst individuals with mild, moderate, and severe hemiplegia. According to research, individuals with severe hemiplegia have strong connection in certain brain areas, which may indicate a compensating mechanism or alternative motor network rearrangement techniques (Guiyuan Cai & Jiayue Xu & Cailing Zhang & Junbo Jiang., 2024). Quinlan E B et al. discovered that patients with more severe injuries had higher iM1 FC in CPMCS after three weeks of standardized robotic therapy, but patients with less severe injuries showed the inverse connection (Erin Burke Quinlan., 2018). These findings may imply that various brain areas are necessary for motor recovery in stroke patients and that additional brain regions may be recruited for functional compensation, which is shown in improved local functional connectivity.

Comments 3: Between the patients selected who suffered a brainstem stroke and healthy patients, was there a significant difference in the resting HbO2 levels?

Response 3: We greatly admire your sensitivity to mixed signals during fNIRS testing. When detecting the cerebral cortex, fNIRS also detects the outer tissue layer of the brain to a certain extent. Variations in oxygenation and blood flow in non-brain tissues may impact fNIRS signaling and cause interference with measurement data. Additionally, brain hemodynamics are also impacted by systemic physiological changes. Examples include: (1) variations in blood pressure, (2) variations in the partial pressure of carbon dioxide (PaCO2), (3) variations in heart rate and vascular tone brought on by interactions between the sympathetic and autonomic nervous systems, and (4) variations in blood flow and oxygenation brought on by head movements, clenching, or raising of the eyebrows. Firstly, we used the long-short source detector separation (SDS) strategy to detect only changes related to brain activity, reducing the confounding effects of strong hemodynamic changes occurring in the extracerebral tissues, and eliminating the possible effects of microcirculation abnormalities on signals caused by the above diseases. At the same time, the Modified Beer-Lambert Law (MBLL) is used to eliminate or reduce the signal deviation caused by non-target factors (such as tissue scattering, instrument noise, physiological interference, other absorbent substances, etc.). In addition, we also strictly followed the enrollment criteria when we included patients. All the included patients with swallowing disorder after brainstem stroke were in stable condition and had no history of serious peripheral artery disease, carotid artery or vertebral artery stenosis, so as to control

---

## [Decision Letter · Decision Letter 1]

15 May 2025

The effects of visual stimulation on the cortical activity of brainstem stroke dysphagia patients: A functional near-infrared spectroscopy study

PONE-D-24-45819R1

Dear Dr. Hui,

We’re pleased to inform you that your manuscript has been judged scientifically suitable for publication and will be formally accepted for publication once it meets all outstanding technical requirements.

Kind regards,

Shashank Shekhar, MD

Academic Editor

PLOS ONE

Additional Editor Comments (optional):

Due to unavailability of second reviewer I reviewed the manuscript and decided based on mine and reviewers 1 input.

Reviewers' comments:

Reviewer's Responses to Questions

**Comments to the Author**

1. If the authors have adequately addressed your comments raised in a previous round of review and you feel that this manuscript is now acceptable for publication, you may indicate that here to bypass the “Comments to the Author” section, enter your conflict of interest statement in the “Confidential to Editor” section, and submit your "Accept" recommendation.

Reviewer #2: All comments have been addressed

2. Is the manuscript technically sound, and do the data support the conclusions?

Reviewer #2: Yes

3. Has the statistical analysis been performed appropriately and rigorously? 

Reviewer #2: Yes

4. Have the authors made all data underlying the findings in their manuscript fully available?

Reviewer #2: Yes

5. Is the manuscript presented in an intelligible fashion and written in standard English?

Reviewer #2: Yes

6. Review Comments to the Author

Reviewer #2: Appreciate the clarifications for the points raised, especially about resting HbO2 levels being accounted for in both patient groups when performing the fNIRS.

7. PLOS authors have the option to publish the peer review history of their article (what does this mean? ). If published, this will include your full peer review and any attached files.

**Do you want your identity to be public for this peer review?** For information about this choice, including consent withdrawal, please see our Privacy Policy .

Reviewer #2: No

---

## [Editor Report · Acceptance letter]

PONE-D-24-45819R1

PLOS ONE

Dear Dr. Hui,

I'm pleased to inform you that your manuscript has been deemed suitable for publication in PLOS ONE. Congratulations! Your manuscript is now being handed over to our production team.

Kind regards,

on behalf of

Dr. Shashank Shekhar

Academic Editor

PLOS ONE